# Patterns of loneliness among high school students: A sociodemographic analysis in Kenya

David M. Ndetei[1,2,3], Andre Sourander[4,5], Christine Musyimi[1,3], Pascalyne Nyamai[1,3], Eric Jeremiah[1,3], Samuel Walusaka[1,3], Victoria Mutiso[1,3], Kirill Vasilchenko[6], Antonio Ventriglio[7], Dinesh Bhugra[8] and Egor Chumakov[9] (iD)

[1]Africa Institute of Mental and Brain Health, Kenya; [2]University of Nairobi, Kenya; [3]World Psychiatric Association Collaborating Centre for Research and Training, Kenya; [4]University of Turku, Finland; [5]Turku University Central Hospital, Finland; [6]Holon Institute of Technology, Israel; [7]University of Foggia, Italy; [8]King's College London, Institute of Psychiatry Psychology & Neuroscience, UK and [9]Saint Petersburg State University, Institute of Medicine, Russian Federation

## Research Article

**Keywords:**
public mental health; childhood experience; geopsychiatry; LMIC

**Corresponding author:**
David M. Ndetei;
Email: dmndetei@uonbi.ac.ke

## Abstract

Loneliness is recognized as a significant public mental health issue, especially among adolescents. There is insufficient research on adolescent loneliness in countries such as Kenya, where adolescents make up 23% of the population. The aim of this study was to examine the prevalence of loneliness among high school students living in different regions of Kenya. This cross-sectional study included 2,652 high school students from ten schools across three Kenyan regions, reflecting both urban and rural settings. Participants completed a questionnaire assessing socio-demographic, educational, and psychological factors, along with their experiences of loneliness over the past year. The level of loneliness was assessed by the question "During the past 12 months, how often have you felt lonely?". Loneliness during the past 12 months (responses "always" and "most of the time") were identified in 17.1% of males and 16.6% of females. Significant factors associated with loneliness included grade level, geographical location, family structure, and perceived economic status. Urban students and those attending schools in Nairobi and Kiambu, as well as those from economically disadvantaged backgrounds, reported higher loneliness. The high prevalence of loneliness highlights the need for targeted interventions, particularly in urban and economically disadvantaged groups.

## Impact statement

This research provides one of the first detailed analyses of loneliness among adolescents in Kenya. This country's mental health system faces significant challenges, including limited infrastructure, low psychiatrist-to-population and other mental health professionals ratios and minimal integration of mental health in primary care. Considering that adolescents represent 23% of the Kenyan population, young people are particularly vulnerable to mental health issues in this country. The study is the first to address the issue of loneliness among adolescents in Kenya and identified key risk factors, such as urban environments, economic disadvantage, specific family structures and a lack of close friends. This evidence is crucial for developing effective, culturally grounded interventions. The study directly informs schools, community leaders and policymakers in Kenya and similar regions on where to target support, for instance, by strengthening peer networks, complementing traditional counseling with proactive and preventative professional interventions and integrating psychosocial care into schools and community programs. Ultimately, this work provides a blueprint for actionable, low-cost strategies to mitigate adolescent loneliness, a known risk factor for more severe mental health issues, thereby contributing to healthier development and better life outcomes for young people in resource-limited settings.

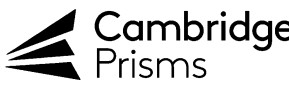



## Introduction

Loneliness is a public mental health issue (Bhugra et al., 2024). It is considered a subjective experience resulting from unmet expectations in social relationships (Akhter-Khan et al., 2024). Conceptually, loneliness is understood as a distressing emotional state (characterized by sadness and emptiness) that arises from a perceived gap between one's desired and actual social relationships (Peplau and Perlman, 1982). Throughout the lifespan, loneliness is particularly significant during adolescence, a critical period characterized by changes in biological, psychological and social development (Luhmann and Hawkley, 2016). This stage of development is also marked by new relationships, exploration of identity and rapid physical growth, all factors possibly leading to a sense of disconnection and isolation (Laursen and Hartl, 2013). Many

studies described an increased loneliness among adolescents in different communities in the last decades (Madsen et al., 2019; Twenge et al., 2021). In a recent meta-analysis exploring the loneliness in 113 countries, it was observed that problematic loneliness was a common experience across all age groups, including adolescents worldwide, with the pooled prevalence for adolescents ranging from 9.2% in South-East Asia to 14.4% in the Eastern Mediterranean region (Surkalim et al., 2022). Loneliness is a social concern in both higher-income countries and lower- and middle-income countries (LMICs) (Akhter-Khan et al., 2024; Amu et al., 2020). Recent research evidence revealed that the overall prevalence of loneliness among adolescents aged 12–15 years from 28 countries in Africa, Asia and the Americas was 10.7% (with boys reporting 8.7%, and girls reporting 12.6%) (Smith et al., 2024). Feelings isolated or alone during adolescence are factors associated with a range of negative consequences, including self-harm, illicit drug use, alcohol consumption, and risky sexual behaviors (Stickley et al., 2014). Also, loneliness is associated with the onset of depression, other common mental health problems and suicidal ideation (Aboagye et al., 2022; Mann et al., 2022). Currently, there is limited understanding of the association between the feeling of loneliness and cultural, economic and sociopolitical factors (Akhter-Khan et al., 2024). Also, differences in the prevalence of loneliness among adolescents in non-Western, low-income nations are poorly described (Rodriguez et al., 2022). Previous findings on children aged 13–17 years from 70 countries across Africa, America and Asia reported that the overall prevalence of loneliness was 11.7%, specifically 18.8% in Kenya (Igami et al., 2023). In 2023, the estimated number of adolescents in Kenya was 13,075,038, representing 23% of the population (UNICEF, 2024).

Kenya's mental health system faces significant challenges, including limited infrastructure, severe shortages of specialist providers (such as psychiatrists [0.19 per 100,000 population]) and an overall scarcity and uneven distribution of other mental health professionals (psychologists, counselors, social workers and psychiatric nurses), particularly at the primary care level (Kumar et al., 2022). Stigma remains pervasive, often rooted in cultural beliefs that associate mental illness with spiritual weakness or moral failure (Ogutu, 2022). These barriers contribute to low service uptake among adolescents. Moreover, mental health remains a low policy priority, with a low budgetary allocation to mental health and a deficiency of trained mental health professionals (Kumar et al., 2021). Kenya's diverse population, encompassing numerous ethnic groups, local languages and a mix of urban and rural settings, presents a rich landscape for research and potential for significant findings. The exploratory design allows culturally specific patterns to surface without imposing a prescriptive predictor framework.

The aim of this study was to examine the prevalence of loneliness and determine its significant associations among adolescents living in different regions of Kenya.

## Methods

### Study participants and procedure

In this cross-sectional study, 2,652 students attending 10 secondary (high) schools from three (Nairobi, Kiambu and Makueni) of 47 regions in Kenya were invited. Of the schools participating in the study, four were located in an urban setting and six were in rural settings. Nairobi, as the capital city, reflects a densely populated and socioeconomically diverse urban environment, while Kiambu and Makueni include semi-rural and rural regions. Schools selected for

this study represented a range of socioeconomic backgrounds. This geographic and institutional diversity aimed to improve the inclusivity of in-school adolescents. All schools included in the study were coeducational. High school education in Kenya is divided into four forms: Form 1, Form 2, Form 3 and Form 4. This system is similar to what some countries refer to as grades or years in high school, but in Kenya (Ndetei et al., 2024), they are called "forms." The age range for the students varies from 13 to 23 years. The data collection was performed between May and June 2022.

All students were randomly assigned to groups of 12–15 students using the permutation block method. The questionnaire (available on request to the first author) included questions about their sociodemographic, educational, linguistic and geographical characteristics, as well as questions with suggested response options on the specific topics of this study. Sensitivity $\chi^2$ tests were repeated with post-stratification weights that mirror the 2023 national sex distribution for secondary-school enrolment (51% boys/49% girls) (Kenya Ministry of Education, 2024).

### Measures

A questionnaire was developed by the authors of the paper, with input from both local experts and a panel of international specialists, and included questions regarding students' demographic (gender and age), educational (high school grade), geographical (location of school), familial and economic characteristics. The level of loneliness was assessed by the question "During the past 12 months, how often have you felt lonely?" This question was previously used in the Global School-based Student Health Survey that was jointly developed by the World Health Organization, the US Centers for Disease Control and Prevention and other UN allies, and in major international studies (Vancampfort et al., 2019). The variable, measured using a five-point Likert scale ("never," "rarely," "sometimes," "most of the time" and "always"), was dichotomized for analysis. Following the methodology of a prior publication, responses of "never," "rarely" and "sometimes" were coded as 0, while "most of the time" and "always" were coded as 1. We also asked about the number of friends and the need for outside help with students' problems, feelings, behavior or emotional trouble. All demographic items were single-choice; subsequent one-hot encoding (OHE) yielded mutually independent binary variables.

### Ethics

All procedures in this work comply with the ethical standards established by the Helsinki Declaration of 1975, as revised in 2013. Kenyatta University Ethical Review Committee approved this study proposal (IRB number – PKU/2456/E1587). A research license was granted from the National Commission for Science, Technology and Innovation (NACOSTI) (license number NACOSTI/P/22/17173). Permission was sought from institutional heads. Written informed consent was obtained from students over 18 years old, while assent and parental permission were obtained from those under 18 years old. The questionnaire was completed by those students who provided informed consent/assent or with the permission of their parents, depending on their age.

### Data analysis

Levels of loneliness were analyzed and their associated factors among demographic, familial and psychological ones. For

continuous variables, the *t*-test or analysis of variance was applied based on group numbers and distribution characteristics. In our statistical analysis, we employed OHE to transform categorical variables into a binary format, which is a common practice for handling categorical data in regression models. This method was chosen because it allows for the inclusion of nominal scale variables without assuming an ordinal relationship, ensuring that our model accurately reflects the independence of categories. For instance, different educational levels and geographical locations, which are nominal variables in our dataset, were transformed using OHE to facilitate their analysis in the context of loneliness. The chi-square ($\chi^2$) test was then utilized to assess the significance of the association between each transformed binary variable and loneliness. Due to the extensive number of tests conducted, the Benjamini–Hochberg (BH) procedure was implemented to control the false discovery rate (FDR), adjusting *p*-values to minimize the likelihood of type I errors. The BH procedure controls the expected false-discovery proportion. This methodological approach supports a more precise and unbiased estimation of the impact of various predictors on loneliness.

Significant variables (BH-adjusted $p \leq 0.05$) were entered into a hierarchical agglomerative cluster analysis (Ward linkage, Euclidean distance on the vector of $\chi^2$ statistics) solely to visualize inter-variable structure; no further variable selection or prediction modeling was performed. Significant variables, identified through the corrected chi-square tests, were subjected to hierarchical clustering using the Ward method, which minimizes within-cluster variance. Euclidean distance between BH-$\chi^2$ statistics was used as the dissimilarity measure; Ward linkage minimized within-cluster variance. Hierarchical clustering was visualized through a dendrogram, utilizing the "elbow" method to determine the optimal number of clusters by noting where additional clusters do not significantly decrease within-cluster variance. The dendrogram served purely descriptive purposes and was not used to decide which variables enter further analyses. Unless otherwise specified, all reported *p*-values are BH-adjusted and evaluated against $\alpha = 0.05$. Statistical data processing was done using the Python programming language with the Pandas, SciKit and Matplotlib libraries.

The direction and magnitude of all bivariate associations between each factor and loneliness are reported. For each factor, we computed category-wise prevalence of loneliness, risk differences, odds ratios (ORs) with 95% confidence intervals (CIs) and Cramér's V as a standardized effect size, alongside Benjamini–Hochberg adjusted *p*-values. Cross-tabulations by loneliness status are presented in Supplementary Table 2B, and full bivariate estimates in Supplementary Table S1.

As a descriptive aid, we present a hierarchical clustering of factors by strength of association with loneliness (Supplementary Figure S1); this visualization is not used for hypothesis testing or inference.

## Results

Two thousand five hundred ninety-six Kenyan high school students responded to the questionnaire (response rate 97.9%) and were included in the study. The sample age ranged from 13 to 23 years old, with a mean age = 16.0 years (standard deviation = 1.38), which is a typical high school age range in Kenya. The wide age range reflects delayed school entry, as many older youth remain in school, still considered adolescents in this context. Table 1 shows students' demographic, educational and geographical characteristics. It is

worth noting that the socioeconomic backgrounds and educational experiences within the sample varied significantly. The distribution of students between the different forms was balanced, with the median level of education being the second form. As expected, we found a higher number of participants from rural areas of Kenya, with the majority of students born in Kenya ($n = 2{,}511$; 96.7%) and a small percentage in other countries. Weighted sensitivity analyses using national sex proportions produced identical significance patterns.

The majority of students ($n = 1740$; 67.0%) reported experiencing feelings of loneliness in the past year (Table 2). Weighted sensitivity analyses using national sex proportions produced virtually identical significance patterns; only three low-prevalence items changed their *p*-value status. Loneliness during the past 12 months (responses "always" and "most of the time") was identified in 17.1% of males and 16.6% of females. More than half of students reported having three or more close friends, reflecting their ability to interact and sociability. In our analysis of factors associated with loneliness, we identified that the need for outside help (defined as the need for assistance from someone outside the immediate family for behavior

**Table 1.** Sociodemographics and educational characteristics of participants

| Characteristics | Category | *n* (%) |
|---|---|---|
| Gender | Male | 1,728 (66.6) |
| | Female | 862 (33.2) |
| | Other | 6 (0.2) |
| Form (high school class level) | 1 | 869 (33.5) |
| | 2 | 729 (28.1) |
| | 3 | 646 (24.9) |
| | 4 | 352 (13.6) |
| Location of school | Rural | 1,602 (61.7) |
| | Urban | 994 (38.3) |
| Parents with whom the respondents were living | Two biological parents | 1,605 (61.8) |
| | Biological mother | 594 (22.9) |
| | Biological father | 62 (2.4) |
| | Step-parent living with the biological parent | 69 (2.7) |
| | Adoptive parents | 23 (0.9) |
| | Foster parents | 10 (0.4) |
| | Grandparent or grandparents | 64 (2.5) |
| | Other relatives (e.g., older sibling, uncle and aunt) | 80 (3.1) |
| | Other | 18 (0.7) |
| | No answer | 71 (2.7) |
| Respondents' assessment of the family's economic well-being in comparison with other families | Not well | 160 (6.2) |
| | Not particularly well | 281 (10.8) |
| | Fairly well | 867 (33.4) |
| | Rather well | 363 (14.0) |
| | Very well | 781 (30.1) |

**Table 2.** Social factors associated with loneliness (overall sample)

| Variable | Male; *n* (%) | Female; *n* (%) | Other; *n* (%) | $\chi^2$ (df); *p*[a] |
|---|---|---|---|---|
| During the past 12 months, how often have you felt lonely? | | | | **$\chi^2$(5) = 12.858; *p* = 0.038** |
| Always | 86 (4.9) | 39 (4.5) | 1 (16.7) | |
| Most of the time | 209 (12.1) | 104 (12.1) | 1 (16.7) | |
| Sometimes | 522 (30.2) | 313 (36.3) | 1 (16.7) | |
| Rarely | 333 (19.3) | 131 (15.2) | 0 (0.0) | |
| Never | 452 (26.2) | 219 (25.4) | 2 (33.3) | |
| No response | 126 (7.3) | 56 (6.5) | 1 (16.7) | |
| How many close friends do you have? | | | | **$\chi^2$(4) = 72.769; *p* = 0.003** |
| 0 friends | 123 (7.1) | 62 (7.2) | 2 (33.3) | |
| 1 friends | 210 (12.2) | 201 (23.3) | 1 (16.7) | |
| 2 friends | 242 (14.0) | 155 (18.0) | 1 (16.7) | |
| 3 or more friends | 1,018 (58.9) | 394 (45.7) | 1 (16.7) | |
| No response | 135 (7.8) | 50 (5.8) | 1 (16.7) | |
| Within the past 6 months, have you at any point felt a need for outside help (someone outside your immediate family) with your problems, feelings, behavior or emotional trouble? | | | | $\chi^2$(3) = 5.028; *p* = 0.169 |
| I have sought outside help | 202 (11.7) | 121 (14.0) | 1 (16.7) | |
| I have considered getting outside help | 597 (34.5) | 275 (31.9) | 2 (33.3) | |
| No, I have not felt the need | 778 (45.0) | 379 (44.0) | 2 (33.3) | |
| No response | 151 (8.7) | 87 (10.1) | 1 (16.7) | |

[a]Differences calculated in comparison between males and females. Statistically significant differences are highlighted in bold.

or emotional issues in the past 6 months) was a relevant factor associated with loneliness in both male and female gender groups (*p* = 0.001). Those who sought help approached friends (*n* = 187; 57.7% of those who responded that they were looking for help; multiple responses allowed), relatives (*n* = 122; 37.7%), religious or spiritual leaders (*n* = 94; 29.0%), teachers (*n* = 85; 26.2%), medical doctors (*n* = 48; 14.8%), psychologists or school counselors (*n* = 45; 13.9%), coaches in sport or other leisure activity (*n* = 34; 10.5%), school nurses (*n* = 6; 1.9%) and others (*n* = 36; 11.1%).

### *Social factors and loneliness*

Significant variations in loneliness were observed across different educational levels, with form 4 showing the highest prevalence of loneliness ($\chi^2$ = 56.937, *p* < 0.001). Age yielded the same $\chi^2$ significance as form and is therefore omitted from further tables. Among geographical factors, we found that living in an urban area was significantly associated with a higher level of loneliness ($\chi^2$ = 37.696, *p* < 0.001) compared to living in a rural area. Those students attending the schools in Nairobi ($\chi^2$ = 37.696, *p* < 0.001) and Kiambu ($\chi^2$ = 24.394, *p* < 0.001) showed more severe levels of loneliness, while those in Makueni ($\chi^2$ = 120.089, *p* < 0.001) showed opposite patterns, suggesting that environmental factors may play a role in the perception of loneliness. Respondents living with two biological parents reported lower levels of loneliness compared to those living with other relatives or non-biological parents ($\chi^2$ = 16.873, *p* = 0.003 for two biological parents; $\chi^2$ = 17.168, *p* = 0.003 for other relatives). Those who perceived their families as economically "Very well" exhibited lower loneliness ($\chi^2$ = 45.698, *p* < 0.001), whereas those who perceived their economic

status as "Not particularly well" reported higher levels of loneliness ($\chi^2$ = 33.486, *p* < 0.001). Those who sought help from psychologists or school counselors reported a higher level of loneliness ($\chi^2$ = 10.999, *p* = 0.038). Individuals with no close friends reported significantly higher levels of loneliness ($\chi^2$ = 77.758, *p* < 0.001), while having two or more friends was associated with reduced loneliness ($\chi^2$ for two friends = 42.417, *p* < 0.001; $\chi^2$ for three or more friends = 74.161, *p* < 0.001). Although the "counselor" variable clusters with wider help-seeking items, its individual $\chi^2$ (Table 2) still indicates a significant link with loneliness. Students from Makueni were significantly *less* lonely than peers in Kiambu or Nairobi ($\chi^2$ = 18.4, *p* < 0.001).

Hierarchical clustering of factors is provided as a descriptive visualization in Supplementary Figure S1 and is not used for inference. The main text focuses on effect sizes and directions summarized in Figure 1 and detailed in Supplementary Table 2B/ Supplementary Tables S1 and S2. The forest plot on Figure 1 encodes both direction (risk vs. protective) and 95% CIs for the reference contrasts.

Points show ORs for reporting loneliness "most of the time/ always" relative to the reference categories; horizontal bars are 95% CIs; the dashed line marks OR = 1. Factors included: sex (ref. Female), location (ref. Urban), county (ref. Nairobi), number of close friends (ref. ≥3) and perceived economic status (ref. Rather/Very well). Estimates come from 2 × K cross-tabulations (bivariate contrasts); *p*-values were obtained from $\chi^2$ tests with Benjamini–Hochberg control of the FDR. Full counts, percentages, Cramér's V and ORs with 95% CIs are reported in Supplementary Table S1.

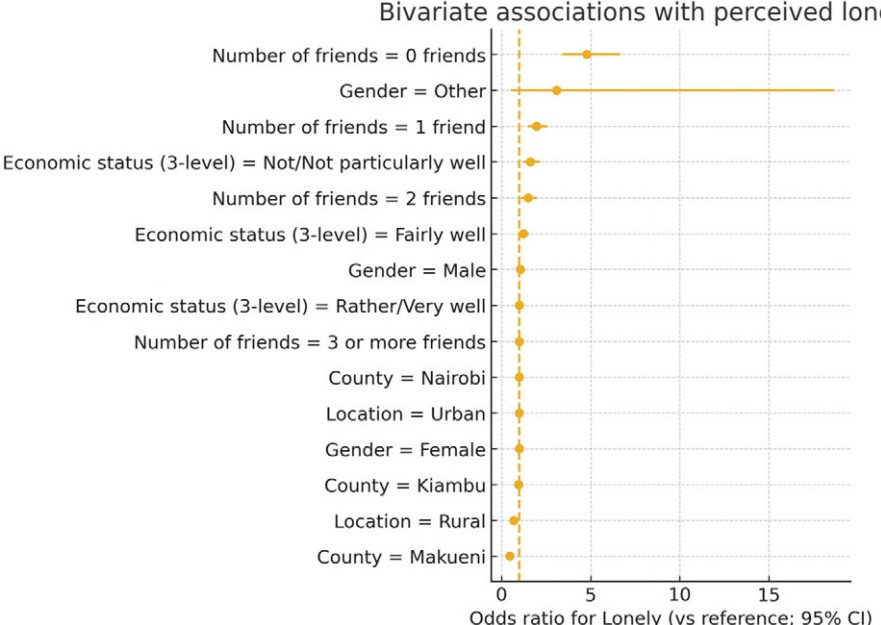

**Figure 1.** Forest plot of bivariate associations with loneliness. Reference categories: ≥3 close friends; Urban; Nairobi; Two biological parents; Rather/Very well. Values correspond to odds ratios with 95% CIs.

Figure 1 shows a clear gradient by number of close friends: relative to having ≥3 friends, odds of loneliness increase stepwise for 2, 1 and 0 friends (OR = 1.48, 1.95, and 4.78, respectively; 95% CIs in Figure 1). County differences are evident, with lower odds of loneliness in Makueni than in Nairobi (OR = 0.46 [0.35–0.60]). Lower perceived economic status is associated with higher odds of loneliness ("Not/Not particularly well" vs. "Rather/Very well": OR = 1.62 [1.22–2.16]). Sex differences are minimal (Male vs. Female OR ≈ 1.05). These bivariate patterns are directionally consistent with the multivariable model (Supplementary Table S2).

Loneliness was coded as 1 for "Most of the time/Always" and 0 otherwise (N = 2,413 non-missing; Lonely = 440, Not lonely = 1,973). The number of close friends showed a graded association with loneliness. Relative to ≥3 close friends, prevalence of loneliness was 18.6% for 2 friends, 23.2% for 1 friend and 42.5% for 0 friends; bivariate ORs were 1.48 [1.10–1.99], 1.95 [1.48–2.58] and 4.78 [3.43–6.67].

In the multivariable generalized linear model with robust (HC3) standard errors, the associations were directionally consistent. Using ≥3 friends as the reference, adjusted ORs (aORs) versus having 0 friends equated to aOR (0 vs. ≥3) = 4.35 [3.14–6.03] (inverse of the ≥3 vs. 0 contrast). Makueni county remained associated with lower odds of loneliness compared to Nairobi (aOR = 0.52 [0.36–0.76]). Living with grandparents and with other relatives showed higher odds of loneliness versus two biological parents (aOR = 3.27 [1.66–6.44] and aOR = 2.66 [1.28–5.53], respectively). Perceived economic status "Not/Not particularly well" trended higher versus Rather/Very well (aOR = 1.28 [0.93–1.77]). Gender differences were small (Male vs. Female aOR ≈ 1.03 [0.74–1.44]).

## Discussion

We explored the associations between various sociodemographic, geographic, linguistic and psychological factors and experiences of loneliness in adolescents living in different regions of Kenya. It was found that some degree of loneliness over the last year was common to 67.0% of respondents, with loneliness reaching 17.1% for males and 16.6% for females. These rates were higher than those reported in the sub-Saharan Africa region (10%) (Aboagye et al., 2022).

Significant factors associated with loneliness included grade-related factors (form 4 students showed the highest prevalence of loneliness), geographical influence (living in the urban settings and schools in Nairobi and Kiambu), family structure (living with non-biological parents or other relatives), perceptions of economic family status as "not particularly well" and social support (lack of close friends). We also found that those students who sought help from psychologists or school counselors reported higher loneliness, suggesting that the need for professional help may correlate with higher levels of loneliness. This finding may also reflect appropriate help-seeking during distress. However, these findings warrant further research.

An earlier study showed that girls, students who experienced bullying victimization and students who reported a lack of close friends were at increased risk of experiencing loneliness (Igami et al., 2023). Evidence of the influence of gender on adolescents' feelings of loneliness in Kenya has also been demonstrated previously, but was not replicated in our study (Rodriguez et al., 2022). Another study in Kenya highlighted the need to strengthen psychosocial support (including mentorship, guidance, coaching and counseling) to foster student adjustment, improve academic outcomes and address loneliness among school learners (Baru et al., 2020).

A recent systematic review of qualitative studies on experiences of loneliness in 15 LMICs (including Kenya) showed that in African studies, loneliness was primarily conceptualized not merely as a lack of social contact (social isolation), but as a painful experience characterized by a sense of rejection and overthinking (Akhter-Khan et al., 2024). Lack of close friends in our study was reported by 7.2% of respondents, with no difference by gender. However, this rate was lower than that previously described in Kenya (13.2%) (Igami et al., 2023). Although more than half of the respondents in

our study reported having three or more friends, 67% still experienced loneliness in the past year. Importantly, the lack of close friends was powerfully associated with loneliness. This suggests that while the subjective experience of loneliness may be complex and qualitative (involving factors such as perceived rejection or rumination), the objective condition of having fewer social contacts remains a significant risk factor among adolescents. Thus, our findings highlight that while the quantity of friendships is a significantly protective factor, it does not fully explain the prevalence of loneliness, underscoring the critical importance of investigating relationship quality and individual expectations in future research.

It is well known that poverty and stigma are common barriers to fulfilling social relationship expectations (Akhter-Khan et al., 2024). Also, some reports described the effect of lower income on feelings of loneliness among adolescents in Kenya (Aboagye et al., 2022). Although we did not assess the objective economic situation of respondents' families, their subjective perceptions of economic status were strongly linked to the experience of loneliness. This study was conducted in Nairobi, Makueni and Kiambu Counties, which include a mix of urban low-income neighborhoods, peri-urban and some rural settings. Many of the participants were drawn from the under-resourced areas where access to quality education, stable employment and mental health resources is very limited. This context supports our findings regarding subjective family's economic well-being, where adolescents' perception of family wealth significantly correlated with loneliness, likely due to social comparison and financial stressors common in low-resource settings.

The availability and type of outside help accessed were significantly associated with loneliness. Individuals who reported a need for outside help with their behavioral or emotional problems in the past 6 months showed a significant level of loneliness. This may suggest that the perceived need for external support may be closely linked to the experience of loneliness. However, in the present study, we cannot conclude the primacy of one or the other variable. A recent review of the mental health interventions for adolescents in sub-Saharan Africa found a low coverage and prioritization of mental health interventions across the region (Mabrouk et al., 2022). This highlighted the critical importance of investing in adolescent mental health promotion initiatives in the region to address the pressing needs of this age group.

Addressing loneliness requires a cultural shift and public sensitization, alongside systemic reforms to enhance service delivery and community-based interventions. These findings can also be applied to school-based adolescent psychosocial support programs in Kenya, suggesting areas that may require focused interventions to alleviate loneliness. Another application of these findings might be the development of targeted social and policy measures to address the underlying predictors of loneliness.

## Limitations

Limitations of the study include the cross-sectional design, and longitudinal data are required to identify reliable predictors of loneliness in the study group. This study highlights loneliness among Kenyan adolescents, focusing on in-school youth. While not generalizable to all adolescents, diverse school types across urban and rural counties offer insights relevant to Kenya's evolving education and youth mental health policies. Also, the wide age range and differences between the experiences of 13-year-olds and 23-year-olds may have affected the results of the study. Limitations of the study may include the noninvolvement of other regions of Kenya and the lack of validated instruments to assess loneliness. However, the large sample size and the relevant number of variables studied (including social and geographical) may add relevance to this study. Furthermore, while this study identifies key correlates of adolescent loneliness, it was not designed to assess the potential influence of educational structures, such as coeducational versus single-sex schooling environments. Future cross-national research that deliberately compares these settings could provide critical insights for targeted, school-based interventions.

## Conclusions

Our findings showed that some grade-related, geographical, familial and social factors were significantly associated with loneliness among high school students in Kenya. While our data indicate that students who sought existing psychological services reported higher loneliness, this correlation underscores a system primarily used reactively. Therefore, our findings suggest the need to move beyond this model and complement traditional counseling with proactive, preventative professional interventions. These interventions should be culturally sensitive, low-cost and suitable for LMICs. Effective "professional help" must include evidence-based, school-embedded programs, such as universal psychoeducation, to destigmatize help-seeking, social-skills training groups to foster connection and systematic screening to facilitate early support. For example, given Kenya's collectivist culture, strategies should focus on strengthening community and family support networks by enhancing community-based adolescent mental health programs, integrating psychosocial content into school curricula, and actively involving caregivers and peers in intervention delivery. Shifting the model toward prevention can help decouple the act of seeking professional support from peaks in loneliness distress and leverage communal structures to promote adolescent well-being.

**Open peer review.** To view the open peer review materials for this article, please visit http://doi.org/10.1017/gmh.2026.10153.

**Supplementary material.** The supplementary material for this article can be found at http://doi.org/10.1017/gmh.2026.10153.

**Data availability statement.** The data that support the findings of this study are available from the first author [DMN] upon reasonable request.

**Author contribution.** D.M.N. conceptualized the study. D.M.N., A.S., C.M., P.S. and V.M. designed the study and collected data. D.M.N., K.V., A.V., D.B. and E.C. provided the formal analysis. D.M.N., K.V. and E.C. prepared the manuscript. A.V. and D.B. supervised the study and writing process. The draft manuscript was reviewed and edited by D.M.N., K.V., A.V. and D.B. All authors made substantial contributions to conception and design, acquisition of data or analysis and interpretation of data, took part in drafting the article or revising it critically for important intellectual content, agreed to submit to the current journal, gave final approval of the version to be published and agree to be accountable for all aspects of the work.

**Financial support.** This research received no specific grant from any funding agency, commercial or not-for-profit sectors.

**Competing interests.** The authors declare none.

**Ethics statement.** This study was approved by the Kenyatta University Ethical Review Committee (IRB number – PKU/2456/E1587).

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
