## [Reviewer Report]

the topic of loneliness and specifically among the adolescents is very important at this time when cases of teenage suicide are increasingly reported. may be you could use your data to analyse the impact off coeducational schools on loneliness and conduct similar studies to other counties to give a national representation. otherwise i acknowledge that you have pointed out this in your paper. keep up good work.

---

## [Reviewer Report]

1. The study addresses a key issue (loneliness) that determines adolescents' well-being and some age-relevant risk factors. A timely topic to study.

2. The statement, “Kenya’s mental health system faces significant challenges, including limited infrastructure, low psychiatrist-to-population ratios (0.19 per 100,000).” What is the place of other mental health professionals, Psychologists, Social workers, Counsellors, etc.?

3. What’s the conceptual definition of loneliness in this study?

4. The statement, “The majority of students (n=1740; 67.0%) reported experiencing feelings of loneliness in the past year (Table 2).” and “More than half of students reported to have 3 or more close friends, reflecting their ability to interact and sociability.” On the other hand, ”while having two or more friends was associated with reduced loneliness (χ² for two friends = 42.417, p < 0.001; χ² for three or more friends = 74.161, p < 0.001).” How do the three statements reconcile?

5. In the conclusion section, there is a statement, “These findings may suggest the need for professional help to address loneliness among youths.” Another statement, “We also found that those students who sought help from psychologists or school counselors reported higher loneliness, suggesting that the need for professional help may correlate with higher levels of loneliness.” How do these statements reconcile? Which “professional help” would work for students, if the one available may correlate with loneliness? Expound or provide distinguishing examples.

---

## [Editor Report]

Dear Authors,

I am pleased to inform you that the independent reviews of your paper are positive, but with a few suggestions and observations that amount to minor revision. I encourage you to address them as you see relevant, and submit the revised draft at the earliest possible.

Best wishes,

Thomas